

# Variations induced by the use of unstable surface do not facilitate motor adaptation to a throwing skill

Francisco J. Moreno[1], David Barbado[1,2], Carla Caballero[1,2], Tomás Urbán[1] and Rafael Sabido[1]

[1] Sports Research Centre/Department of Sport Sciences, Miguel Hernández University of Elche, Alicante, Spain
[2] Institute for Health and Biomedical Research (ISABIAL Foundation), Miguel Hernández University of Elche, Alicante, Spain

## ABSTRACT

Induced variability by the use of unstable surfaces has been proposed to enhance proprioceptive control to deal with perturbations in the support base better. However, there is a lack of evidence about its benefits facilitating motor adaptions in upper body skills. In this experiment, practice on an unstable surface was applied to analyze the adaptations in an upper limb precision throwing skill. After a pretest, twenty-one participants were randomly allocated into two groups: one group practiced the throwing task on a stable surface and the other group practiced the same task on an unstable support base. Differences in throwing performance between pre- and post-practice were analyzed in accuracy, hand movement kinematics and variability of the throw in both surface conditions. Fuzzy entropy of the horizontal force was calculated to assess the complexity dynamics of postural sway. Participants improved their performance on the stable and the unstable surface. Induced variability using an unstable surface reduced participants' variability and the complexity of postural sway, but it did not facilitate a superior adaptation of the throwing task. The results suggest that the variations induced by unstable surfaces would fall far from the family of specific motor solutions and would not facilitate additional motor performance of the throwing task.

## INTRODUCTION

Motor variability has been traditionally understood as a movement inaccuracy caused by the noisy neuromuscular function (*Churchland, Afshar & Shenoy, 2006*; *Harris & Wolpert, 1998*; *Osborne, Lisberger & Bialek, 2005*; *Schmidt et al., 1979*; *Shmuelof, Krakauer & Mazzoni, 2012*), and, thus, it must be minimized to promote better performance. Nevertheless, in recent years, some findings suggest that motor variability can also be driven by the central nervous system to play a functional role in adaptive motor behaviors (*Churchland, Afshar & Shenoy, 2006*; *Galea et al., 2013*; *Mandelblat-Cerf, Paz & Vaadia, 2009*; *Pekny, Izawa & Shadmehr, 2015*; *Sober, Wohlgemuth & Brainard, 2008*; *Tumer & Brainard, 2007*; *Wu et al., 2014*). According to this rationale, human motor variability would promote these adaptive behaviors by facilitating motor adjustments that refine

Corresponding author
David Barbado, dbarbado@umh.es

movement performance during the interaction with changing environments (*Barbado et al., 2017*; *Davids et al., 2003*). Nonlinear analysis of motor variability has been frequently used to address the functional variations revealed by human movement (*Barbado et al., 2012*; *Goldberger, Peng & Lipsitz, 2002*). Specifically, non-linear tools like entropy measures (*Barbado et al., 2012*; *Manor et al., 2010*) or detrended fluctuation analysis (DFA) (*Amoud et al., 2007*; *Barbado et al., 2017*; *Wang & Yang, 2012*; *Zhou et al., 2013*) have been used to quantify the predictability and autocorrelation of the time series, which has been related to the system mechanisms for movement or postural control.

Based on the functional perspective of motor variability, a large number of studies have been focused on incorporating variable practice conditions to facilitate motor adaptations (see *Caballero et al. (2017)* for a review); however, scientific literature has also found contradictory findings that question the usefulness of variability in practice to foster motor learning in closed skills (*Edwards & Hodges, 2012*; *Johnson & McCabe, 1982*; *Pigott & Shapiro, 1984*; *Wrisberg & Mead, 1981*; *Wrisberg & Mead, 1983*; *Zipp & Gentile, 2010*). These controversial results suggest that the optimal load of motor variability should be induced and modulated through the manipulation of the practice conditions according to the individual's characteristics and the task constraints (*Caballero et al., 2017*). *Ranganathan & Newell (2013)* proposed a framework to examine the effect of induced variability in practice and distinguished two levels in the introduction of induced-variability: task goal and execution redundancy. In the first level, practitioners would induce variations on the task goal with the intention of causing different task outcomes that will improve the generalization to other task variations. In the second level, variations are intended to cause the same outcome with different movements, exploring the redundancy of the motor system. Ranganathan and Newell considered that the manipulation of practice variability at the execution redundancy level would encourage the exploration of multiple solutions to perform a task, improving the flexibility of the motor system. This flexibility would be related to the ability to perform motion adjustments to refine the movement dealing with subtle changes in the body or in the environment.

One of the potential ways to improve the system's flexibility to reach a specific task goal would be to increase movement fluctuations by enhancing the task balance demands. In the rehabilitation and sports fields, practitioners frequently use unstable surfaces (*e.g.*, foam cushions, wobble board, inflated rubber disc) to increase postural control demands while throwing, catching, or kicking a ball (*Hrysomallis, Buttifant & Buckley, 2006*; *Kisner & Colby, 2007*). From the perspective of functional motor variability, the potential learning benefits caused by modifying the support base features during the above-mentioned tasks would rely on the stimulation of the individuals' ability to perform postural adjustments, which, in turn, would allow these individuals to adapt to the possible variations that might occur while performing the task. This rationale would be supported by previous balance studies that used textured insoles, which found that manipulating the support base features improved motor control by altering sensorimotor inputs *via* mechanoreceptors on the plantar surface of the feet (*Hatton et al., 2012*), enhanced human joints self-perception (*Waddington & Adams, 2003*), and enhanced movement discrimination (*Steinberg et al., 2016*; *Steinberg et al., 2017*). In addition, an EMG study has shown that both anticipatory

and compensatory postural adjustments are adapted while catching a ball in an unstable standing posture (*Scariot et al., 2016*). Based on this rationale, increasing motor variability during practice through the manipulation of the support base could enable highly functional behavior, revealing useful sources of information to regulate the movement (*Davids et al., 2004*). However, as far as the authors know, little empirical evidence supports the usefulness of inducing motor fluctuations through unstable surface to promote faster learning in throwing (*Fisek & Agopyan, 2021*; *Zacharakis et al., 2020*) or related tasks as shooting (*Aydin & Revans, 2019*; *Hung et al., 2021*), while other works do not find any additional benefits (*Alicia Nian, 2017*; *Caballero, Luis & Sabido, 2012*; *Sillero et al., 2022*).

Thus, this study aimed to offer highlights on the topic of using unstable surfaces to improve motor control adaptations in a throwing task. In the experiment, practice on surface instability was applied to analyze the adaptation to an upper limb precision throwing skill. Additionally, nonlinear analysis of the variability was used to assess how practicing on an unstable surface affects the fluctuations in the participants' output movement. The complexity of postural sway parameters could be related to adaptive behaviors in response to the perturbations caused by the unstable base of support. Finally, it was hypothesized that the group that trained in the unstable condition would obtain larger throwing performance improvements than the group that trained in the stable condition, especially when throwing accuracy was evaluated in the unstable condition.

## MATERIALS & METHODS

### Participants

Twenty-one participants (four female and 17 males, age = 30,39 ± 5,82 years) took part in this study. All the participants were right-handed and participated voluntarily in the study. The data were treated anonymously, and all participants were informed of the risks and benefits of the study and signed a written consent according to the ethical guidelines of the Miguel Hernández University of Elche (IRB Approval: 2013.83.E.OEP).

### Experimental setup and procedure

The participants were asked to perform a test which consisted in throwing a tennis ball with their right hand to hit a target located on a wall which was at 1.65 m of distance and 2.15 m high. The throws were performed in a standing position with their feet placed at shoulder width, and their foot orientation was such that the vector formed by their heels was situated in a parallel position to the mediolateral axis (Fig. 1). Their left-hand was resting on their hip during the entire test.

All the participants performed a pretest consisting in a serial throwing task during four 30-s sets with a 30-s rest between sets. Two sets were performed on a stable surface (floor) and two sets on an unstable surface (standard BOSU balance trainer). Afterwards, participants were randomly assigned to one of the two experimental groups to carry out 10 practice series of the 30-s duration throwing task with a 30-s rest period between series. All the series had to be carried out by the participants on the same day. One group practiced in the stable condition (on-floor training) and the other group on the unstable surface (on-BOSU training). After the practice, the participants performed a post-test in the same
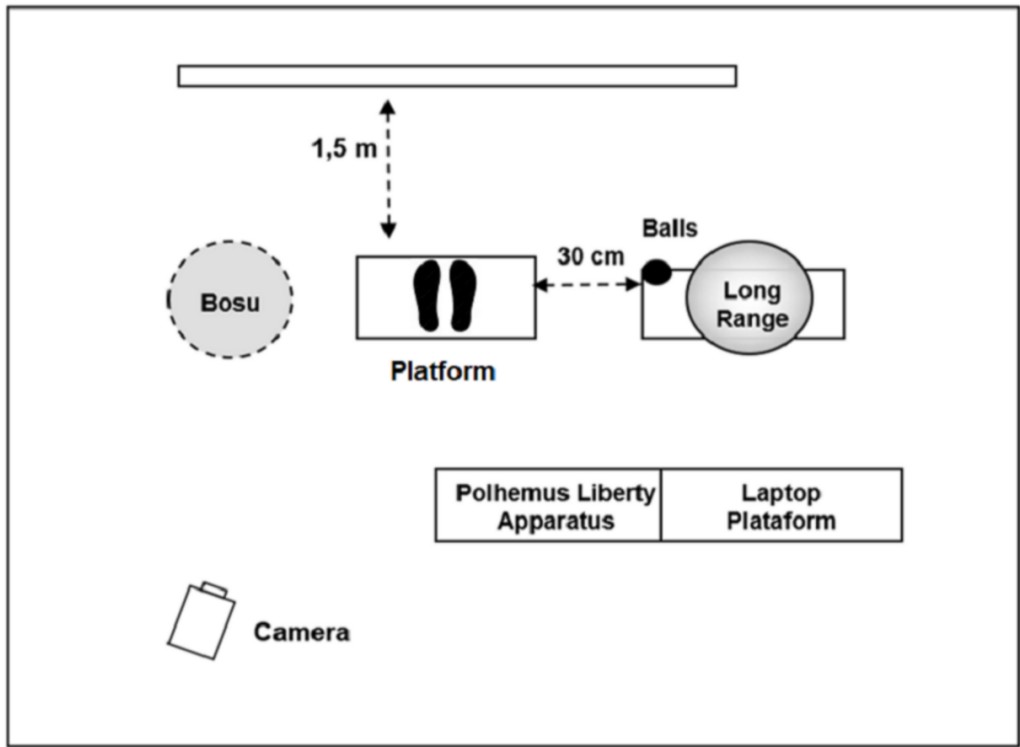

**Figure 1  Experimental setup.** Representation of the experimental setup in the laboratory.

conditions as the pretest. In order to avoid progressive error, stable and unstable conditions in pretest and post-test were counterbalanced.

Kinematic variables of the throwing hand movements were recorded through a three-dimensional Polhemus Liberty electromagnetic position tracking system (Polhemus Ltd., Colchester, VT, USA) at 240 Hz sampling frequency. This system has an accuracy of .076 cm for the mediolateral (ML), antero-posterior (AP) and vertical (V) axis position and .15° for angular orientation (Azimuth, Elevation and Roll). A sensor was placed on the back of the dominant hand (metacarpus medial part).

During the task, the horizontal forces in the base of support in the AP and ML axis were recorded using a force platform (Model 9286AA; Kistler, Winterthur, Switzerland) at 240 Hz sampling frequency during the throws.

The impacts of the ball were video recorded with a "Sony HDR-SR8E" digital camera (Sony, Tokyo, Japan) (50 Hz sampling frequency) to establish the impact zone of each throw. The bounces of the ball were digitalized with Kinovea® 0.8.15 (http://www.kinovea.org/) to calculate the accuracy of the throws. A Matlab 7.11 (Mathworks, Natick, MA, USA) routine was used for the calculation of real-space Cartesian coordinates of the ball bounces.

## Data analysis and reduction

The magnitude of error was calculated through mean radial error (MRE), measured as the average of the absolute distance of the balls bounce with respect to the center of the target. Kinematics of the hand were analyzed by measuring the position of the hand in the three axes (ML, AP and V) given by the position tracking system described above and calculating the instantaneous resultant velocity, computing both variables at the time of the release of the ball. The overall kinematic variability of the hand exhibited by the participants during the series of throws was calculated through the standard deviation of the mean of the instantaneous resultant velocity of the hand at the time of the release of the ball.

Horizontal force values in the AP and ML axis during the task were extracted. No filtering was carried out on the data as filtering can affect the nonlinear results (*Kyvelidou et al., 2010*). Fuzzy entropy (FE) was calculated (*Chen et al., 2007*) to assess the complexity of the time series of horizontal forces in the AP and ML axis. FE typically returns values that indicate the degree of irregularity in the signal. Higher values of FE thus represent greater irregularity in the time domain of the signal. Lower values represent greater repeatability of vectors and are a marker of lower irregularity in the signal output. High and low entropy scores have been respectively associated with greater or lower flexibility to perform movement adjustments to adapt to intrinsic or extrinsic perturbations (*Barbado et al., 2012*; *Manor et al., 2010*). The following parameter values were used: vector length, $m = 2$; tolerance window, $r = 0.2 \times$ SD; and gradient, $n = 2$.

## Statistics analysis

To compare the effects of both throwing conditions of practice on the outcome measures and complexity variables, a mixed repeated-measures ANOVAs with two within-group factors, *time* (two levels: pre- and posttest) and *test condition* (two levels: throwing on the floor and throwing on the BOSU), and one between-group factor, *training type* (two levels: on-floor training and on-BOSU training) was carried out. A Shapiro–Wilk analysis was carried out to evaluate the normality of the distribution of data. The alpha value of significance effect was set at $p < 0.05$. Bonferroni adjustments to the $p$ values were applied for post-hoc multiple comparisons. The effect size was also calculated using the partial eta squared ($\eta p2$) to provide the proportion of the overall variance that is attributable to the factor. Effect size values above 0.64 were considered strong, values below 0.64 and above 0.04 were considered moderate and $\leq 0.04$ were considered small (*Ferguson, 2016*). In addition, Hedges' $g$ index was used to estimate the effect size of each pair comparison using the standard deviation of the change between repeated measure conditions. Hedges' $g$ scores were categorized as: trivial ($g < 0.2$), small ($0.2 \leq g < 0.5$), moderate ($0.5 \leq g < 0.8$) and large ($g \geq 0.8$). Data for all the treatments were performed using the statistical package IBM SPSS 25.

## RESULTS

Participants from both groups showed less accurate throws in the unstable test condition (on BOSU) than in the stable test condition (on floor) independently from when they were tested (*test condition* main effect; $F_{1,19} = 16.648$, $p = 0.001$, $\eta p2 = 0.467$). Throwing

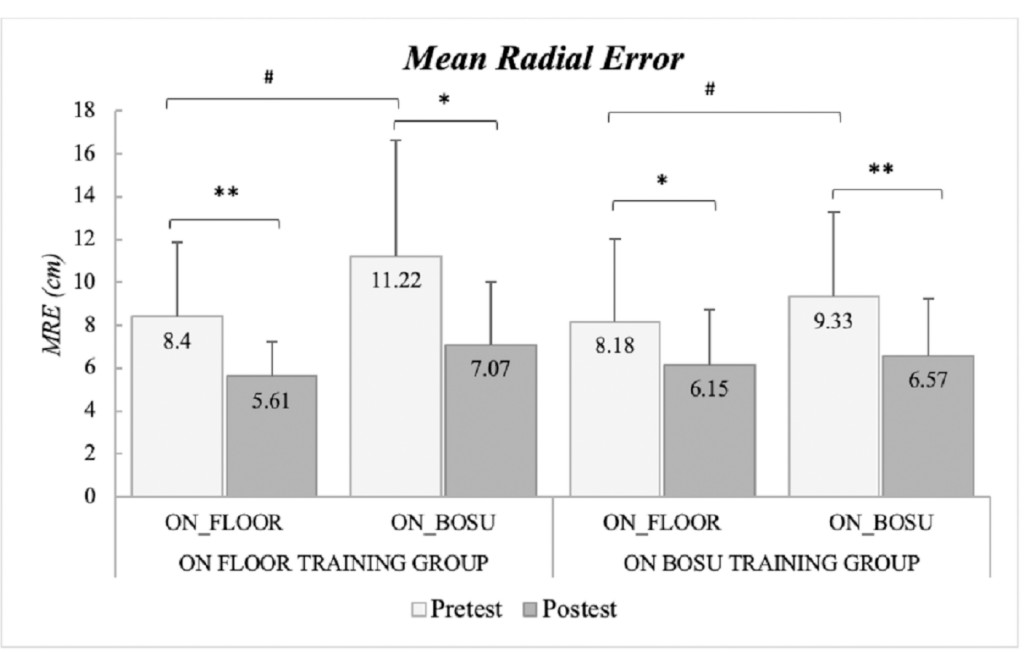

**Figure 2 Mean radial error (mean ± SD) in the pre-test and post-test according to the throwing surface and the practice group.** *Significant differences from pre-test ($p < 0.05$). **Significant differences from pre-test ($p < 0.001$). #Significant differences from pre-test ($p < 0.05$).

accuracy improved significantly after the training period showing lower MRE in the post-test (*time* main effect; $F_{1,19} = 24.193$, $p < 0.001$, $\eta p2 = 0.560$). There were no interaction effects between *training type* and *time*. In the post-test, the reduction of the error in the throws on the BOSU was higher than those performed on the floor post-test (*time ×test condition* interaction effect; $F_{1,19} = 4.999$, $p = 0.038$, $\eta p2 = 0.208$) and the differences between the two throwing conditions were reduced in the post-test, although they were non-significant (Fig. 2; Supplementary Material S1). There were no interaction effects according to the type of practice.

Regarding hand kinematics, no differences were found in the hand position at the time of the ball release between the two throwing situations (on-floor *vs* on-BOSU) in the pretest nor in the posttest. Practice did not change the hand position either. Analyzing hand velocity, the participants exhibited lower hand velocity in the throws on the BOSU than those performed on the floor (*test condition* main effect; $F_{1,19} = 17.465$, $p = 0.001$, $\eta p2 = 0.479$). After practice, the velocity of the hand decreased significantly only in the on-floor throwing condition and only for the on-floor training group (Table 1).

The analysis of the hand variability at the time of the ball release showed no differences between throwing on-BOSU and on-floor in the pretest, nor in the post-test. With practice, there was not a significant reduction of hand variability for any group but an interaction effect was found (*time × training type* interaction effect; $F_{1,19} = 5.998$, $p = 0.018$, $\eta p2 = 0.263$). The pair comparison revealed that only the on-BOSU training

**Table 1  Velocity of hand and kinematic hand variability in pre-test and post-test according to the throwing surface and the practice group.** Units in m/s.

| | Hand velocity | | | Hand variability | | |
|---|---|---|---|---|---|---|
| | Pretest | Posttest | $t$ ($p$) effect size[b] | Pretest | Posttest | $t$ ($p$) effect size[b] |
| | | | On-floor training group ($n = 9$) | | | |
| Throwing on the FLOOR | $4.28 \pm 0.34$ | $4.15 \pm 0.21$ | 2.438 (0.041) 0.774 | $0.13 \pm 0.05$ | $0.15 \pm 0.04$ | $-1.021$ (0.337) |
| Throwing on the BOSU | $4.15 \pm 0.39$ | $4.10 \pm 0.20$ | 0.677 (0.517) 0.215 | $0.15 \pm 0.02$ | $0.15 \pm 0.03$ | 0.232 (0.822) |
| $t$($p$) | 2.753 (0.025) | 2.456 (0.040) | | $-1.230$ (0.254) | 0.294 (0.777) | |
| Effect size[a] | 0.874 | 0.780 | | $-0.391$ | 0.093 | |
| | | | On-bosu training group ($n = 12$) | | | |
| Throwing on the FLOOR | $4.28 \pm 0.21$ | $4.25 \pm 0.20$ | 0.753 (0.467) 0.210 | $0.14 \pm 0.04$ | $0.12 \pm 0.03$ | 1.936 (0.079) 0.540 |
| Throwing on the BOSU | $4.24 \pm 0.21$ | $4.20 \pm 0.25$ | 0.714 (0.490) 0.199 | $0.16 \pm 0.07$ | $0.12 \pm 0.04$ | 2.578 (0.026) 0.718 |
| $t$($p$) | 1.474 (0.169) | 2.176 (0.052) | | $-0.920$ (0.377) | $-0.695$ (0.501) | |
| Effect size[a] | 0.411 | 0.606 | | $-0.256$ | $-0.194$ | |

Notes.

Student $t$ tests for repeated measures comparing between test conditions (throwing on the floor *vs.* throwing on the BOSU) of pre- and post-intervention tests.

[a] $t$, $p$, and effect size values refer to pair comparisons between test conditions (throwing on the floor *vs.* throwing on the BOSU).

[b] $t$, $p$, and effect size values refer to pair comparisons between pre- and post-intervention tests.

$p$ significant values were adjusted to $<0.025$ by Bonferroni correction.

Hedges' $g$ index was used to estimate the effect size of each pair comparison using the standard deviation of the change between repeated measure conditions.

group showed a reduction in variability on both throwing surfaces. The on-floor training group did not reduce their variability after practice (Table 1).

Participants from both groups showed higher FE (more complexity) in the medial-lateral (*test condition* main effect; $F_{1,19} = 12.909$, $p = 0.002$, $\eta p2 = 0.381$) and anterior-posterior forces (*test condition* main effect; $F_{1,19} = 16.907$, $p < 0.001$, $\eta p2 = 0.446$) when they threw in the BOSU condition compared to the floor condition in the pretest and post-test (Table 2). In addition, interaction effects between *training type*, *time* and *test condition* were observed in FE values of medial-lateral forces ($F_{1,19} = 5.030$, $p = 0.037$, $\eta p2 = 0.209$) and anterior-posterior ($F_{1,19} = 5.422$, $p = 0.031$, $\eta p2 = 0.222$). The pairwise comparison revealed that no differences were found between groups after practice in FE values. The on-floor training group did not show any significant change in FE values of force fluctuations. Nevertheless, the on-BOSU training group decreased complexity (lower FE) of force fluctuations in the posttest in the throws on-BOSU in medial-lateral axis and anterior-posterior axis (Table 2).

## DISCUSSION

The aim of this study was to offer highlights on the using unstable surfaces to improve motor control adaptations. For that, instability on the base of support was applied while training an upper limb precision throwing skill. The effect of practicing in the unstable test condition was compared with practicing in a stable test condition. Within participant pre-post practice differences were analyzed in the accuracy, kinematics and variability of

**Table 2 Fuzzy entropy of data forces on the base of support in medial-lateral (ML) and anterior-posterior (AP) axes according to the practice group and throwing surface.**

| | Fuzzy entropy in the AP axis | | | Fuzzy entropy in the ML axis | | |
|---|---|---|---|---|---|---|
| | Pretest | Posttest | $t(p)$ effect size[b] | Pretest | Posttest | $t(p)$ effect size[b] |
| | | On-floor training group ($n = 9$) | | | | |
| Throwing on the FLOOR | $0.68 \pm 0.22$ | $0.62 \pm 0.18$ | 1.555 (0.158) 0.494 | $0.69 \pm 0.29$ | $0.66 \pm 0.24$ | 1.095 (0.305) 0.348 |
| Throwing on the BOSU | $0.91 \pm 0.24$ | $0.86 \pm 0.17$ | 0.529 (0.611) 0.168 | $0.99 \pm 0.34$ | $0.98 \pm 0.33$ | 0.110 (0.915) 0.035 |
| $t(p)$ | −1.569 (0.155) | −3.230 (0.012) | | −1.710 (0.126) | −2.603 (0.031) | |
| Effect size[a] | −0.498 | −1.025 | | −0.543 | −0.826 | |
| | | On-bosu training group ($n = 12$) | | | | |
| Throwing on the FLOOR | $0.68 \pm 0.17$ | $0.79 \pm 0.32$ | −1.803 (0.099) −0.502 | $0.70 \pm 0.25$ | $0.83 \pm 0.36$ | −1.501 (0.162) −0.418 |
| Throwing on the BOSU | $1.05 \pm 0.24$ | $0.89 \pm 0.17$ | 3.543 (0.005) 0.987 | $1.16 \pm 0.29$ | $0.96 \pm 0.25$ | 3.324 (0.007) 0.926 |
| $t(p)$ | −4.185 (0.002) | −1.101 (0.294) | | −3.602 (0.004) | −0.952 (0.362) | |
| Effect size[a] | −1.166 | −0.307 | | −1.004 | −0.265 | |

Notes.

Student $t$ tests for repeated measures comparing between test conditions (throwing on the floor *vs.* throwing on the BOSU) of pre- and post-intervention tests.

[a] $t$, $p$, and effect size values refer to pair comparisons between test conditions (throwing on the floor *vs.* throwing on the BOSU).

[b] $t$, $p$, and effect size values refer to pair comparisons between pre- and post-intervention tests.

$p$ significant values were adjusted to $< 0.025$ by Bonferroni correction.

Hedges' $g$ index was used to estimate the effect size of each pair comparison using the standard deviation of the change between repeated measure conditions.

the throws. Additionally, nonlinear analysis of the horizontal force fluctuations in the base of support was applied to explore changes in movement dynamics in relation to the ability to adapt to unbalance situations.

As expected, in general terms, the throws on the stable base of support condition were more accurate than in the unstable test condition. This fact supports the idea that performing an accuracy task on an unstable platform caused a decline of the motor control of the main task, this is to say, of the throw. Overall, when the surface was unstable (on the BOSU), participants also threw slower (lower hand velocity) which could be related to an adaptive motor control strategy (*Behm et al., 2015*). The dynamics of the horizontal forces in the base of support are interpreted in terms of the resultant output of the whole movement coordination during the throws. The FE values of force fluctuations revealed more complexity in the unstable surface test condition, both in the ML and the AP axes. Increased movement complexity has been related to a more functional exploration of the information provided by the environment (*Davids et al., 2003*), and even to more voluntary or intentional motor control (*Urbán et al., 2019*; *Van Orden, Kloos & Wallot, 2011*). Previous studies that applied unstable platforms while performing discrete motor patterns, like handball throwing (*Urbán, Gutiérrez & Moreno, 2015*), suggested higher flexibility in the motor patterns and better adaptations in movement coordination in order to improve motor control (*Behm & Colado, 2012*). Based on this rationale, it was hypothesized that the more complex motor variability induced by the unstable support would lead to a better adaption to the specific task demands.

After training, all the participants improved their throwing accuracy, independent of the training group (on-BOSU or on-floor). Contrary to what was hypothesized, the group that trained in the stable condition (the on-floor training group), obtained a very good motor adaptation when throwing on the stable base of support but also on the unstable base of support. The group which trained in the unstable condition(on-BOSU training) also improved the accuracy of the throws mainly in the specific conditions they trained, and to a lesser extent in the on-floor condition (exhibiting the lowest mean differences pre-post). Contrary to other findings, in which induced variability in practice has been shown to be more useful than constant practice to improve performance and transference (*Hinkel-Lipsker & Hahn, 2017*; *Leving et al., 2016*), our results support that, in this case, increased variability in practice did not seem to support improved performance above a traditional practice setting to improve throwing accuracy. Nevertheless, it must be pointed out that variable practice did not hamper throwing accuracy compared to regular practice, which would indicate that individuals adapted their behavior to cope with the induced perturbations. This seem to be supported by the fact that only the on-BOSU training group reduced hand variability in both situations in the post-test while no changes were observed for those participants who practiced in the stable condition. That could suggest that participants who trained on the unstable base of support adapted their movement to minimize the variability induced by the instability. However, they did not achieve better adaptation to increase the accuracy in any of the two throwing conditions compared to the on-floor training group. Considering that our protocol only tested the short-term adaptations after practice, future studies should check if reduced hand variability caused by practicing on unstable surfaces could induce throwing improvements after long-duration training programs.

Regarding the changes in the dynamics of the horizontal force fluctuation after training, we have only found changes caused by practice in the group that trained on the unstable surface. Despite the fact that the participants showed higher complexity in the force fluctuations when they threw on the unstable surface, the participants who trained in that unstable condition reduced their complexity. The lower autocorrelation (*i.e.,* higher DFA values) observed in the postural sway parameters has been previously interpreted as an index of a lower number of postural adjustments (*Barbado et al., 2017*). Therefore, this on-BOSU training group showed an adaptative behavior in response to the fluctuations caused by the unstable base of support during practice. Additionally, this adaption has been observed mainly in the specific task dimension in which the task was constrained (*Moreno, Caballero & Barbado, 2022*; *Urbán et al., 2019*). In this sense, instability in the base of support during practice within appropriate ranges could cause stress on the system, but in turn it should promote motor control adaptations.

In summary, the results indicated that instability in the base of support improved motor skill adaption in the throwing as much as a regular practice. In fact, this type of practice did not change postural control adjustments that could lead to better performance. From the authors' point of view, two interpretations can explain these results. On the one hand, previous researchers have highlighted that to improve motor adaptions, it is always necessary to adjust the amount of variability. *Renshaw et al. (2016)* suggested that

a low amount of variability in practice would not promote additional motor pattern forming or an adequate system reorganization, but too much variability would make the environment unmanageable for the individual. In this sense, the unstable surface could entail excessive postural sway fluctuations for the participants of our study (*Davids et al., 2004*), which resulted in higher levels of stress on the system (*Moreno & Ordoño, 2015*). Conversely, previous studies that have reported benefits from the use of unstable surfaces were conducted with young or inexperienced athletes (*Aydin & Revans, 2019*; *Fisek & Agopyan, 2021*; *Zacharakis et al., 2020*), except for *Caballero, Luis & Sabido*'s study (*2012*). Furthermore, studies conducted with experienced athletes have not found additional improvements caused by training on unstable surfaces (*Alicia Nian, 2017*; *Sillero et al., 2022*), except for *Hung et al.*'s study (*2021*), which did not perform a group comparison. Therefore, although both, the level of the athlete and the practice load could be crucial factors on the effect of variable practice on motor adaptation, their role and how they interact are not clear yet. A significant limitation for testing this hypothesis is related to the fact that, currently, there is no index available that can reflect the impact of a practice training load on an individual. Therefore, future studies should look for measurable parameters to quantify the internal load imposed by different practice schedules.

On the other hand, the instability in the base of support, which could theoretically induce variability in the movement redundancy level (*Ranganathan & Newell, 2013*), could be so unspecific and cause unexpected effects in the adaption to the practice. It has been proposed that variable practice should be understood with vector properties, considering not only the magnitude but the orientation (*Moreno & Ordoño, 2015*). This could indicate that, in order to produce the desired adaptation, the most appropriate magnitude and range of variation should be established considering the task characteristics. Variable practice should be directed at enabling variations in the performance that allow to solve the task within the "goal-equivalent manifold" (*Cusumano & Cesari, 2006*) or around the redundant task space of elemental variables (good variance) (*Latash, Scholz & Schöner, 2002*; *Scholz & Schöner, 1999*). Variations far from the family of solutions that solve the task would not facilitate better motor performance and may even cause unwanted adaptations. However, movement redundancy analyses could not be performed because no additional information about hand or ball trajectory was collected. Therefore, new studies must be designed to address the specific relationship between the use of unstable surface-based training programs and the individuals' movement redundancy.

Finally, it must be pointed out that, in our protocol, only acute adaptations after practice were assessed. Thus, future studies should implement long-term training programs to elucidate in which cases inducing variability through unstable surface-based practice does or does not lead to greater improvements in throwing accuracy than regular practice.

## CONCLUSIONS

Variability induced by using an unstable surface facilitated adaptations in movement coordination or improved motor control as much as traditional practice in the throwing task measured in this experiment. The highly functional exploration provided by redundant

variability seems to be specific to the sources of information needed to regulate the movement. Finally, it has to be noted that this experiment has focused on the immediate motor adaption instead of on the consolidation effects or transference. Future research should test the mid- or long-term effects of the use of unstable platforms to improve motor control of secondary motor skills that have to deal with perturbations in the base of support like catches, throws or hits. However, these results do provide insight into the acute effect of practice on unstable surfaces and how motor performance is affected.

### Funding

This study was financially supported by the Economy, Industry and Competitiveness Ministry of Spain, project cod DEP2016- 79395-P, Spanish Government. The funders had no role in study design, data collection and analysis, decision to publish, or preparation of the manuscript.

### Grant Disclosures

The following grant information was disclosed by the authors:
Economy, Industry and Competitiveness Ministry of Spain, project cod: DEP2016-, 79395-P, Spanish Government.

### Competing Interests

The authors declare there are no competing interests.

### Author Contributions

- Francisco J. Moreno conceived and designed the experiments, analyzed the data, prepared figures and/or tables, authored or reviewed drafts of the article, and approved the final draft.
- David Barbado conceived and designed the experiments, analyzed the data, authored or reviewed drafts of the article, and approved the final draft.
- Carla Caballero conceived and designed the experiments, performed the experiments, analyzed the data, authored or reviewed drafts of the article, and approved the final draft.
- Tomás Urbán conceived and designed the experiments, performed the experiments, analyzed the data, prepared figures and/or tables, authored or reviewed drafts of the article, and approved the final draft.
- Rafael Sabido conceived and designed the experiments, analyzed the data, authored or reviewed drafts of the article, and approved the final draft.

### Human Ethics

The following information was supplied relating to ethical approvals (*i.e.*, approving body and any reference numbers):
The Miguel Hernández University of Elche granted Ethical approval to carry out the study within its facilites (2013.83.E.OEP).

## Data Availability

The raw measurements are available in the Supplemental Files.

## Supplemental Information

Supplemental information for this article can be found online at http://dx.doi.org/10.7717/peerj.14434#supplemental-information.

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
