# Peer review of "Variations induced by the use of unstable surface do not facilitate motor adaptation to a throwing skill"

_PeerJ, doi:10.7717/peerj.14434_

## Round 0.1 · original submission · Major Revisions

There are some relatively large differences of opinion between the two reviewers, so I am happy to provide you the opportunity to respond to both reviewers' comments. Make sure you look to address these comments, especially those from reviewer one if you wish to have this manuscript accepted for publication in PeerJ.

·

Basic reporting

The paper is reasonably written. There are, nonetheless, some issues in terms of sentences and aggregation of too much information in few sentences.
The literature referenced seems to be based only on papers that advocate that variability (and variability of practice) is a feature that must be emphasized. I do not agree with such choice and would like to invite the authors to read more on problems of variability (and variability of practice) on Cardis et al. (2018 - Journal of Neurophysiology); Newell (2003 - Research Quarterly for Exercise and Sport); for instance.
Additionally, the authors should be a little more cautious on advocating (or assuming) that all variability is exploration (or leads to exploration) as it just can refer to instability in the movement (see, for instance, Kelso, 1995). Not all variability leads to better outcomes and variability, per se, cannot be equated to exploration. There are types of exploration and only some might lead to finding solutions (please consider our paper Pacheco et al., 2020, Ecological Psychology).
Despite the aforementioned issues, I did not understand how unstable surfaces would support exploration or improvement more than a constant practice. The authors do not explain it in sufficient detail.
The figures and tables are ok. I did not get the raw data from the review files.

Specific:
- Lines 49-52: I suggest the authors to consider reading Vaillancourt & Newell (2003, Journal of Applied Physiology). These authors demonstrate that higher indices of complexity do not result in better performances. In fact, this seems to depend on task constraints.
- Lines 53-55: The sentence needs rewriting. Either "motor variability drives the exploration" or "The functional role of motor variability is to drive"
- Lines 63-66: Despite the argument of Ranganathan & Newell (2013), it is not true that all individuals (when in a constant pracitce) exploit redundancy. See Pacheco & Newell (2015, human movement science) on an example of how individuals largely vary in between.
- Lines 66-68: I am unsure whether Wu et al. used variability of practice. As far as I am aware, they just found that highly variable individuals improved faster.
- Lines 70-89: There seems to be a lot of content to unpack here and I did not understand the logic behind it. 1)Increasing "information differentiation" (or discrimination) does not need to result from variability - and I did not see how it would. 2) "Creation of more information opportunities during the performance" is hard to comprehend. Please, unpack the idea. 3) How unstable surfaces lead to better anything seems still nebulous from the text. Not sure there is convincing evidence for it. 4) Finally, citing speculative papers from the "Ecological Dynamics" does not provide the logic that would put together, unstable surfaces, exploration (that is not the same as variability), motor coordination and control.
- Lines 93-106: I do not see the justification for the study. 1) Variations in implements, movements, or orientation seem to differ, by large, from unstable surfaces. I suggest the authors to provide a direct logical link between the latter and the skill. 2) The previous study had "no very conclusive results". Why is that? How the current paper would solve the issue. Is there any issue from the previous paper? 3) I did not find the rationale for the hypothesis. Please expand on that. 4) How the complexity variable can help on the paper? There are more specific measures of exploration in the literature (see Pacheco et al., 2021; Ecological Psychology)

Experimental design

As stated in the previous section of the review, I did not understand the rationale of the paper. If the case is just for questioning if more variability is better for learning, I think the question does not address any identified knowledge gap. This is even more the case as the authors have a manuscript stating that variability of practice seems to depend on the task (a discussion largely unexplored here).

Specifics:
- Practice was measured in time? I did not see the number of trials, just ten practice series. Was it based on time? Did the participants performed the same amount of trials?
- To understand variability, movement redundancy and so on, analyses on the release parameters should be made (as performed elsewhere, see Tommasino et al., 2021, Plos One, or Muller & Sternad, 2004, Journal of Motor Behavior).

Validity of the findings

My main issue on the validity of the findings refer to the use of variables that do not relate to exploration or redundancy to discuss about these issues. Otherwise, I have no comments.

Reviewer 2 ·

Basic reporting

While for the most part, the writing is clear, there are some instances of phrases or words used that should be reconsidered to aid clarity.
The introduction is good, but I would like to see a short section on fuzzy entropy and the use that has to help researchers understand more about postural control.
Lines 50 51: the references provided here do not support the claim. Please find other references that do support that claim. Suggestion provided in the annotations in the document reviewed.
The paper is largely well referenced, but, there are some instances in which some references don’t support the statements made, that is, they are not relevant. Suggestion provided in the annotations in the document reviewed. These need to be attended to before publication can be considered.
The document conforms to the Journal’s standard and discipline norm style.
The figures and table are well presented and described adequately.
Raw data was supplied.

Experimental design

The research is an original piece of work and fits within the scope of the journal.
The research questions is clearly stated and the authors show clearly where they see their research filling a gap in current knowledge.
Data was collected in an appropriate way and ethical considerations were detailed.
The methods used to collect data were clear and would enable replication. The dependent variables need to be described a little more clearly (Esp. lines 149 – 154). Position of the hand at ball release was not presented in the results. In particular, variability of the hand is not clearly described. (also see the comment in the annotation line 151-152).

Validity of the findings

The authors state that to their knowledge a study like the current experiment has not been completed before. The rationale for the study was clearly stated and the benefit to the research area was well addressed.
Data are robust and appropriate statistical techniques have been used to avoid errors.
While the discussion and conclusions are good, there are instances where clarity could be enhanced by re-writing. For example, the authors (line 258) make a statement that could be expanded upon. Their results showed that not only does constant practice led to improvements in performance, but the unstable surface practice group also improved their performance, so it could be said that practice on an unstable surface is beneficial too.
In line 270, I think the authors need to reword their statement. The authors cited don’t say that fewer (“lower number”) postural adjustments are indicated by lower entropy.
I was also expecting to see a comparison of Fuzzy Entropy between groups post-practice (i.e., did unstable practice group have different FE post practice compared to stable practice group?). This may have shed some light on the effectiveness of unstable surface practice, that is, did it enable a different mode of control (i.e., different fuzzy entropy) compared to stable practice. I also expected to see a between group comparison of the size of the changes made (to dependent variables) between phases of the study.

Additional comments

The authors have presented an interesting study in which two groups of participants practiced a throwing accuracy task while standing on an unstable surface (group 1), or on the floor (group 2). This is a well-designed study that allowed for a comparison of throwing performance and upper limb kinematics between pre-practice and post practice. The authors used a conventional accuracy measurement to determine performance and a combination of traditional and contemporary statistics to describe the kinematics contributing to performance. The results showed that while performance did improve for both groups under both stable and unstable throwing conditions, changes to the kinematic variables measured did not provide much evidence of changes to motor behaviour to accommodate to unstable throwing conditions. The authors provide a good discussion of the main results and some useful conclusions are presented.

Annotated reviews are not available for download in order to protect the identity of reviewers who chose to remain anonymous.

---

## Round 0.2 · Minor Revisions

The two reviewers and I see many improvements in this manuscript. I think you still need to look critically at the comments of reviewer one and better address them if you wish this paper to be accepted for publication in PeerJ.

·

Basic reporting

No comment.

Experimental design

As acknowledged by the authors, functional variability (or exploration of the goal space) cannot be directly addressed in the paper. Despite the authors bringing the idea of complexity measures being somewhat related to exploration, what seems to come out from the response to the reviewers (I might be wrong) is that they agree with me that complexity measures and their interpretation are dependent on the task constraints.
I still do not see a proper explanation of how the current paper aims to solve the discrepancy of the previous papers cited (Fisek & Agopyan, 2021; Zacharakis et al., 2020; Aydin & Revan, 2019; Hung et al., 2021; Alicia-Nian, 2017; Caballero et al., 2012; Sillero et al., 2022). In fact, the authors do not even compare or discuss these papers in their discussion. I would suggest that beyond of the rationale linking postural adjustments to throwing, the paper should explain why some papers do and others do not find the "expected" results then.

Validity of the findings

The authors should add the methodological issue (mentioned in the Experimental Design section) as limitation to the study.

Reviewer 2 ·

Basic reporting

The amendments made to the original manuscript have enhanced the paper.
One reference is not appropriate and needs amending.
* In the tracked changes document, line 206, I don't think the Kyvelidou et al., 2009 paper cited in text and listed in the reference list is the correct paper. I think it should be:
Kyvelidou, A., Harbourne, R. T., Shostrom, V. K., & Stergiou, N. (2010). Reliability of center of pressure measures for assessing the development of sitting postural control in infants with or at risk of cerebral palsy. Archives of physical medicine and rehabilitation, 91(10), 1593-1601.

Experimental design

I am happy with the amendments made to the methods.

Validity of the findings

I am happy with the amendments made to the discussion section.

---

## Round 0.3 · Minor Revisions

The two reviewers and I thank you for your amendments to the previous version. A small number of relatively minor revisions are still required before this can be accepted for publication.

Reviewer 2 ·

Basic reporting

No further comment

Experimental design

No further comment

Validity of the findings

No further comment

Additional comments

There are a few minor issues that need to be addressed. See review document.

Annotated reviews are not available for download in order to protect the identity of reviewers who chose to remain anonymous.

---

## Round 0.4 · accepted · Accept

Thank you for attending to all the reviewers' comments. I am now happy to recommend this paper be accepted for publication in PeerJ.